

# Interactions of semiconductor Cd-based quantum dots and Cd²⁺ with gut bacteria isolated from wild *Salmo trutta* fry

Renata Butrimienė[1], Agnė Kalnaitytė[2], Emilija Januškaitė[2], Saulius Bagdonas[2], Živilė Jurgelėnė[1], Dalius Butkauskas[1], Tomas Virbickas[1], Danguolė Montvydienė[1], Nijolė Kazlauskienė[1] and Vesta Skrodenytė-Arbačiauskienė[1]

[1] Institute of Ecology, Nature Research Centre, Vilnius, Lithuania
[2] Laser Research Center, Physics Faculty, Vilnius University, Vilnius, Lithuania

Corresponding author
Vesta Skrodenytė-Arbačiauskienė,
vesta.skrodenyte@gamtc.lt

## ABSTRACT

**Background:** With the rapid development of nanotechnology, more and more nanoproducts are being released into the environment where they may both pose ecological risks and be toxic to living organisms. The ecotoxicological impact of quantum dots (QDs), a class of nanoparticles (NPs), on aquatic organisms is becoming an emerging issue, this due to their nano-specific properties, to the physico-chemical transformation in the environment and to the possible release of toxic metals from their structure such as Cd.

**Methods:** In this work, (i) spectroscopic measurements of commercially available Cd-based QDs (CdSe/ZnS-COOH) were made at various pH values (5.0 and 7.0) to study their interactions (at a concentration of 4 nm) with various strains of Gram-positive and Gram-negative gut bacteria after short-term exposure and (ii) the antibacterial efficacy of QDs and Cd²⁺ (at a concentration 0.09–3.56 mM) against gut bacteria isolated from wild freshwater *Salmo trutta* fry was studied at different temperatures (15 °C and 25 °C) and pH values (5.0 and 7.0) by applying a well-established disc diffusion assay.

**Results:** Twenty-six gut bacterial isolates from wild *Salmo trutta* fry were identified as *Aeromonas* spp., *A. popoffii*, *A. salmonicida*, *A. sobria*, *Carnobacterium maltaromaticum*, *Buttiauxella* sp., *Listeria* sp., *Microbacterium* sp., *Shewanella putrefaciens* and *Serratia* sp. Cd-based (CdSe/ZnS-COOH) QDs at a concentration of 4 nm were found to be stable in aqueous media (with pH 7.0) or starting to form aggregates (at pH 5.0), thus, apparently, did not release heavy metals (HMs) into the media over 48 h in conditions of light or dark and did not show antibacterial efficacy on the gut bacteria isolated from wild *Salmo trutta* fry after short-term (9 h and 48 h) incubations. Cd²⁺ was found to produce significant dose-dependent toxic effects on bacterial growth, and the size of the inhibition zones on some of the tested strains significantly correlated with temperature. The most sensitive and the most resistant to Cd²⁺ were the Gram-positive bacteria, for which the minimum inhibitory concentration (MIC) values of Cd²⁺ were 0.09–0.27 mM and 3.11–3.29 mM respectively and varied significantly between the tested temperatures (15 °C and 25 °C). The MIC values of Cd²⁺ for the Gram-negative bacteria (18 out of 22 strains) ranged from 0.44 to 0.71 mM and did not differ significantly between the tested temperatures. Among the selected Gram-positive and Gram-negative strains, those

with the higher sensitivity towards Cd$^{2+}$ also revealed relatively stronger signals of QDs photoluminescence (PL) when transferred after incubation into fresh medium without QDs. In addition, the formation of endogenous metalloporphyrins observed spectroscopically in some bacterial strains indicates certain differences in metabolic activity that may play a protective role against potential oxidative damage.

## INTRODUCTION

With the rapid development of nanotechnology, more and more nanoproducts (NP) are being released into the environment where they may both pose ecological risks and be toxic to living organisms (*Ma et al., 2018*; *Sahoo et al., 2021*). The results of earlier studies have demonstrated that ingested NPs could bioaccumulate in the gut and cause acute toxicity for aquatic organisms (*Ma et al., 2018*; *Asghari et al., 2012*). Quantum dots (QDs) are a class of NPs with numerous biomedical applications such as drug delivery, live imaging and medical diagnosis, as well as having use in the electronics industry (*Hardman, 2006*; *Wagner et al., 2019*; *Cotta, 2020*; *Abdellatif et al., 2022*). The cores of QDs consist of diverse metal complexes (*e.g.*, semiconductors, noble metals and magnetic transition metals) and are usually encapsulated by a shell (*Hardman, 2006*; *Matea et al., 2017*). According to a review by *Rocha et al. (2017)*, the concentration of QDs in the aquatic environment is unknown. *Balde et al. (2017)* stated that only 20% of electronic waste is properly recycled and a significant number of manufactured displays have QDs cores containing Cd. Additionally, *Bechu et al. (2021)* noted that the release of QDs into the environment occurs through leakages and discharges from engineered landfills and unregulated waste dumps into the soil and groundwater, as well as from unregulated recycling in unsafe conditions. It can be stated that as the demand for QDs and production volumes further increase, their release into the environment will only increase. Due to their nano-specific properties and physico-chemical transformation, the ecotoxicological impact of QDs on aquatic organisms is an emerging problem, resulting in the possible release of toxic metals such as Cd from the QDs structures (*Rocha et al., 2015a*, *2017*; *Santana et al., 2015*; *Rotomskis et al., 2018*). Previous studies have identified leakage of toxic ions, such as Cd, from QDs and the generation of reactive oxygen species (ROS) as the main contributing factors to QDs cytotoxicity (*Chen et al., 2012*; *Green & Howman, 2005*; *Matos et al., 2020*). The degradation of QDs is dependent on the NPs type and surrounding environment factors, such as pH and salinity (*Mahendra et al., 2009*; *Domingos, Franco & Pinheiro, 2013*; *Rocha et al., 2015b*, *2017*). In addition, a recent study has shown that the toxicity of QDs to organisms depends on their ability to aggregate (*Rotomskis et al., 2018*; *Jurgelėnė et al., 2021*).

The effect of QDs on the gut microbiota of aquatic organisms is an emerging and relatively insufficiently explored research area. The gut microbiome becomes a central theme in environmental toxicology, because, in addition to its role in chemical

detoxification, the microbes within interact with the host immune system and they play a key role in modulating the host's physiology (*Adamovsky et al., 2018*; *Butt & Volkoff, 2019*; *Perry et al., 2020*). Currently, only a few studies have quantified the effects of NPs on the gut microbiome (*Merrifield et al., 2013*; *Adamovsky et al., 2018*; *Evariste et al., 2019*; *Yang et al., 2021*; *Chen et al., 2022*). Furthermore, relatively little known is currently known about bacterial cell interactions with QDs. Previous studies have shown that QDs exhibit antibacterial activity depending on their structure and functionalization (*Lu et al., 2008*; *Patra et al., 2014*; *Rajendiran et al., 2019*). The antimicrobial effects of QDs mainly manifest themselves *via* three molecular mechanisms: (i) the destruction of cell walls/cell membranes (the QDs interact with the phospholipid bilayer, thereby roughening and shrinking the cell membrane; the cell can also be ruptured due to the direct attachment of the QDs, this causing the discharge of cellular components); (ii) the production of ROS, which destroy cells; and (iii) through binding with nucleic material (DNA/RNA), thus inhibiting cell proliferation (*Rajendiran et al., 2019*). *Mahendra et al. (2009)* and *Arshad et al. (2016)* have shown that CdSe/ZnS core-shell QDs maintain an almost stable structure, are not toxic and do not exhibit any antibacterial properties against Gram-positive (*Bacillus subtilis*) and Gram-negative bacteria (*Escherichia* coli and *Vibrio harveyi*) at near-neutral pH. Studies of *Mir et al. (2018)* revealed that ZnSe/ZnS core-shell QDs demonstrated antibacterial activity against Gram-positive (*Staphylococcus aureus*) bacteria, the zone of inhibition was detected at pH 7.4. However, no such activity was observed against Gram-negative (*E. coli*) strain. In another study, a colony-forming capability assay showed that CdSe QDs with a ZnS shell can induce small effects on bacterial viability, whereas CdSe QDs without shells exhibited significant dose dependent toxic effects on the bacteria *Shewanella oneidensis* MR-1 and the *Bacillus subtilis* SB 491 (*Pramanik et al., 2018*). A similar trend was shown by *Lu et al. (2008)* that the CdTe QDs without shells can effectively kill *E. coli* in a concentration-dependent manner. *Monrás et al. (2014)* revealed that CdTe-GSH QDs with 5 nm diameter display higher toxicity than QDs with 3 nm diameter. QDs mainly induced expression of genes involved with $Cd^{2+}$ stress, while $Te^{2-}$ did not contribute significantly to QDs-mediated toxicity since cells incorporate low levels of tellurium. Release of $Cd^{2+}$ in the case of 5 nm QDs was higher, it also induced genes related to oxidative stress response and membrane proteins.

Previously, it was found that short-term (24 h and 96 h) exposure to Cd-based (CdSe/ZnS-COOH) QDs at a concentration of 4 nm affects the functions of the cardiorespiratory system of rainbow trout larvae and embryos. Cd-based QDs accumulated and got stuck in the chorion of the fish embryos, while they accumulated and distributed in the region of the gills in the case of the fish larvae (*Rotomskis et al., 2018*; *Jurgelėnė et al., 2021*). Based on these published results, a QDs concentration of 4 nm was chosen to study the interaction of Cd-based QDs with gut bacteria isolated from wild *Salmo trutta* fry in the current study. Thus, the toxicity of QDs is affected by two very different phenomena: dissolution and aggregation.

It has been found that low pH values can affect the stability of QDs and release the HMs from the cores/shells of the QDs (*Domingos, Franco & Pinheiro, 2013*). The structural integrity of QDs is reflected by the stability of the spectroscopic properties. It is known that

the core size of the QDs determines the spectral position of the PL peak and that characteristic spectral changes indicate a reduction in particle size (*Kuo et al., 2008*). The release of Cd ions can only take place after significant damage to the shell structure of the QDs, which leads to drastic spectroscopic changes (*Karabanovas et al., 2009*). *Mahendra et al. (2009)* found that QDs were potentially safe materials at near-neutral pH, but, in contrast, weathering of various types of QDs under acidic (pH < or = 4) or alkaline (pH > or = 10) conditions significantly increased bactericidal activity due to the rapid release of Cd and Se ions following QDs destabilization upon loss of the organic coating. In fish stomachs, pH values are within the acidic range. The stomach acid working in all monogastric animals is HCl, a very strong inorganic acid produced by gastric glands (parietal cells) (*Solovyev et al., 2015*). Data previously published on freshwater salmonid juveniles indicate that the postprandial pH in the gastric lumen can be arrested at values between 4.0 and 5.0 for at least 8 h (*Bucking & Wood, 2009*; *Bravo et al., 2018*). Temperature was one of the most important factors affecting pH values in fish guts: the pH values in fish guts were higher in the cold seasons (average water temperature 5–10 °C) than in the summer seasons (water temperature 22–25 °C) (*Solovyev et al., 2018*). Thus, the aims of this study were to evaluate (i) the antibacterial efficacy of Cd-based QDs (CdSe/ZnS-COOH, at a concentration of 4 nm) and $Cd^{2+}$ (at a concentration 0.09–3.56 mM) against gut bacteria isolated from wild freshwater *Salmo trutta* fry at different temperatures (15 °C and 25 °C) and pH values (5.0 and 7.0); and (ii) to examine the interactions of Cd-based QDs with various strains of Gram-positive and Gram-negative gut bacteria after short-term exposure at pH 5.0.

## MATERIALS AND METHODS

### Materials
In this study commercially available semiconductor Cd-based CdSe/Zns core/shell QDs (cat. No. A10200; Life Technologies, Carlsbad, CA, USA) were used. The QDs were coated with a polymer layer with carboxylic acid (−COOH) side groups that allow facile dispersion of QDs and got a negative surface charge in aqueous solutions with retention of their optical properties. The certified standard $Cd^{2+}$ (1,000 mg/L) in $HNO_3$ (0.5 mol/L, Certipur® solution) was purchased from Merck (Darmstadt, Germany).

### Spectroscopic measurements of QDs
The stability of the spectral properties of CdSe/ZnS-COOH QDs were measured in phosphate $KH_2PO_4$ –NaOH buffer solution (PBS) (0.05 M) with different pH values (pH 5.0 and pH 7.0) in the dark at room temperature (about 18 °C). Some samples at pH 7.0 were constantly illuminated over 48 h using a white fluorescent lamp (11W/827; Osram Duluxstar, Taiwan, China) from a distance of about 25 cm. The pH 5.0 samples were made from PBS pH 7.0 acidified with HCl (12 M). The concentration of QDs in the stock solution was 8 μM. The 1 ml volumes of QDs samples were prepared by diluting the stock solutions with PBS to a main final concentration of 4 nm. The measurements of the spectra of the QDs samples were made in a 10 × 4 mm quartz cuvette (Hellma, Jena, Germany) 1 h after the beginning of the experiments, the during 24 h and after 48 h.
The absorbance spectra were measured using an AvaSpec-2048 fiber optic (Avantes, Apeldoorn, The Netherlands) absorption spectrometer, while a LS55 spectrofluorimeter (PerkinElmer, Waltham, MA, USA) was used to register PL spectra. An excitation wavelength for PL was set at 405 nm, the excitation slit was 10.0 nm, and the emission slit was 2.5 nm. The CdSe/ZnS-COOH QDs possessed a strong PL in the red spectral region, with a PL peak of 620 nm, and the initial full width at half maximum (FWHM) was about 25 nm in the PBS buffer solutions with pH 7.0 (Fig. S1). OriginPro9 (OriginLab, Northampton, MA, USA) software was used for data analysis.

## Isolation and identification of wild *Salmo trutta* fry gut bacteria

Wild *Salmo trutta* fry specimens were collected from the Kena River (54.650756, 25.633395) by electrofishing in early autumn 2020. The water temperature in the Kena River ranges from 2.5 °C to 18 °C. The collected fish specimens were euthanized by a blow to the head following international, national, and institutional guidelines for the care and use of animals (Directive 2010/63/EU; LT 61-13-005). Fish specimens were aseptically dissected and the gut contents were squeezed out. Serial dilutions of homogenates were prepared in sterile PBS (pH 7.0, Oxoid Ltd, Hampshire, UK) (up to $10^{-6}$), from which 100 μL aliquots were spread onto Tryptone Soya Agar (TSA, Oxoid Ltd, Hampshire, UK) plates. The plates were then incubated at 20 °C for 48–72 h. Twenty-six morphologically different colonies were randomly picked and re-streaked on TSA plates. Bacterial genomic DNA extraction and molecular identification of isolates were performed as previously described (*Skrodenyte-Arbaciauskiene et al., 2012*). Polymerase chain reaction (PCR) products 1,500 bp were sequenced using an automated ABI 310 DNA Sequencer and the Big Dye Terminator Cycle Sequencing Ready Reaction Kit (Perkin Elmer, Waltham, MA, USA). The NCBI Basic Local Alignment Search Tool (BLAST; http://www.ncbi.nlm.nih.gov/) and Ribosomal database project classifier (RDP; http://rdp.cme.msu.edu/) were used to assess similarity between the obtained 16S rRNA gene 1,500 bp sequences and sequences contained within the nucleotide database.

The 16S rRNA gene sequences were deposited in the DDBJ/EMBL/GenBank databases under the following accession numbers: MW441229–MW441233; OL739143–OL739153.

## Antibacterial assay by disc-diffusion method

The antibacterial effect was studied by disk-diffusion assay according to *Mir et al. (2018)* and *Travlou et al. (2018)*. The gut bacterial cultures were grown overnight on Tryptone Soya Broth (TSB, Oxoid Ltd, Hampshire, UK) until reaching the optical density of 1.0 ± 0.1 at 600 nm, corresponding to $10^8$ colony forming units (CFU) per ml. A 100 mL aliquot of the bacterial culture was evenly spread on a TSA agar plate. Sterile paper discs (6 mM diameter, Whatman, Maidstone, UK) impregnated with 10 μL of $Cd^{2+}$ (at the concentration 0.09–3.56 mM or 10–400 mg/L) and Cd-based QDs suspensions (at the concentration of 4 nm) (pH 7.0) were placed on these bacterial spread plates (4–5 paper disk replicates of the same concentration). QDs and $Cd^{2+}$ samples were prepared by diluting stock solutions with PBS (pH 7.0) to the final concentration. Blank samples were sterile discs impregnated with sterile PBS solution (pH 7.0). The plates were incubated at

15 °C and 25 °C for 48 h in natural light (combined day/night) and in the dark. The experiment was repeated with Cd-based QDs at pH 5.0. The Cd-based QDs samples were prepared by diluting the stock solution with PBS (pH 5.0) to the final concentration of 4 nm. PBS, TSA, and TSB were acidified with HCl to pH 5.0. HCl was used because parietal cells in the mucosa secrete HCl into the fish's stomach. After incubation, the Petri dishes were photographed using a smart-phone camera. Images were produced under UV light. The diameters of the growth inhibition zone were measured using the ImageJ software image analyzer and expressed as mm (mean ± SD). MIC values were assessed by determining the lowest concentration causing bacterial growth inhibition.

## Interactions of Cd-based QDs with bacteria: spectroscopic measurements after 9 h exposure

### Bacterial incubation with QDs

The bacterial cultures were grown overnight on TSB at 25 °C by shaking on an incubator shaker (Stuart SI500) at 250 rpm until an optical density of $1.0 \pm 0.1$ at 600 nm was reached that correlated to the log phase of growth. The bacterial solutions were centrifuged at 7,500 g for 15 min, and the pellets washed twice with sterile PBS (pH 5.0). After discarding the supernatant, the bacterial pellets were resuspended and diluted with PBS (pH 5.0) to a density of $\sim 2 \times 10^8$ colony forming units per mL (CFU/mL) and then a Cd-based QDs stock solution was added to obtain the concentration of 4 nm. The Cd-based QDs-exposed bacterial cells were incubated on a rotary shaker at 150 rpm (Stuart SI500) for 9 h at 25 °C in natural light. Samples of bacterial cultures, which were grown under the same conditions and prepared in the same way, but without the addition of the Cd-based QDs, served as corresponding controls.

For spectroscopic measurements, the following samples were prepared: (i) bacterial culture with supernatant (PBS, pH 5.0) (control); (ii) bacterial culture plus Cd-based QDs with supernatant (PBS, pH 5.0); (iii) bacterial culture plus Cd-based QDs without supernatant, in this case bacterial pellets were resuspended in fresh PBS (pH 5.0); (iv) supernatant. Each treatment was replicated three times. The measurements of the fluorescence spectra of the samples were made in a $10 \times 4$ mm plastic cuvettes (Sigma Aldrich, St. Louis, Germany) using a LS55 spectrofluorimeter (PerkinElmer, Waltham, MA, USA). Excitation wavelengths for fluorescence were set at 400 and 440 nm, the excitation and emission slits were 5 nm.

### Bacterial colony counting

The count of bacteria in the samples after 9 h exposure to Cd-based QDs and those of a control group without QDs were evaluated by the spread plate method. Ten-fold serial dilutions of the tested cultures were made in PBS (pH 5.0) and 100 μl of the dilutions (up to $10^{-7}$) were spread on TSA plates in triplicate. The plates were incubated at 25 °C for 48 h and the CFU/mL were calculated afterwards and expressed as mean ± SD.

## Ethics approval

All animal work was conducted on public land and waterways, it did not involve protected, threatened or endangered species, and complied with relevant national and international guidelines and legislation. A permit to perform sampling of *Salmo trutta* fry specimens in the Kena River, Permit No. 025, was issued in 2020 to the Nature Research Centre by the Environmental Protection Agency under the Ministry of the Environment of the Republic of Lithuania. All applicable international, national and institutional guidelines for the care and use of animals were followed (Directive 2010/63/EU; LT 61-13-005).

## Statistical analysis

For statistical evaluation, data on the MIC zone were first checked for normal distribution using Kolmogorov–Smirnov and Shapiro–Wilk tests. A two-way ANOVA (F-statistics, $p < 0.05$) followed by a Post-hoc Tukey's HSD test was used for data. STATISTICA (7.0 Software, Inc., Exton, PA, USA) was used to calculate the statistics.

# RESULTS

## The spectroscopic properties of QDs

The spectroscopic properties of QDs were investigated to determine the structural stability of the QDs under the applied experimental conditions. The initial optical density of CdSe/ZnS-COOH QDs was found to be higher at pH 5.0 than at pH 7.0 (Fig. 1A). In contrast, the initial PL intensity of QDs was lower at pH 5.0 and higher at pH 7.0 (Fig. 1B). The PL spectrum of QDs had a peak at about 620 nm in PBS with pH 7.0, but a bathochromic shift of about two nanometres was observed in the more acidic solution (Fig. 1A, insert). Thus, as expected, the CdSe/ZnS-COOH QDs demonstrated better initial spectroscopic properties (a higher intensity of PL and a bigger quantum yield) in PBS with pH 7.0 than in pH 5.0 1 h after preparation of samples. The decreased PL intensity with the bathochromic shift in the acidic media can be associated with the tendency towards stronger aggregation, which occurs in QDs samples with lower pH.

After 1 day in the dark at room temperature, the PL intensity of Cd-based QDs decreased by about 25% from the initial value in the samples with pH 7.0 and by about 60% with pH 5.0 (Fig. 2A). The peak PL intensities of all QDs samples decreased over 2 days, though a faster decrease was observed in solutions with pH 5.0. During this period of time, not only was a decrease in intensity of the PL band observed in the QDs samples, but also a progressing bathochromic shift under illumination at pH 7.0 for 48 h (Fig. 2B). However, this shift could not reach the degree that was registered immediately in more acidic media. On the other hand, the spectral position of the PL peak at pH 5.0 remained stable during the experiment time. Thus, the difference in pH of the samples caused stronger effects on the optical properties of CdSe/ZnS-COOH QDs than environmental lighting conditions over 48 h, although it seems that the same mechanisms of QDs surface modification were predominant in both cases.

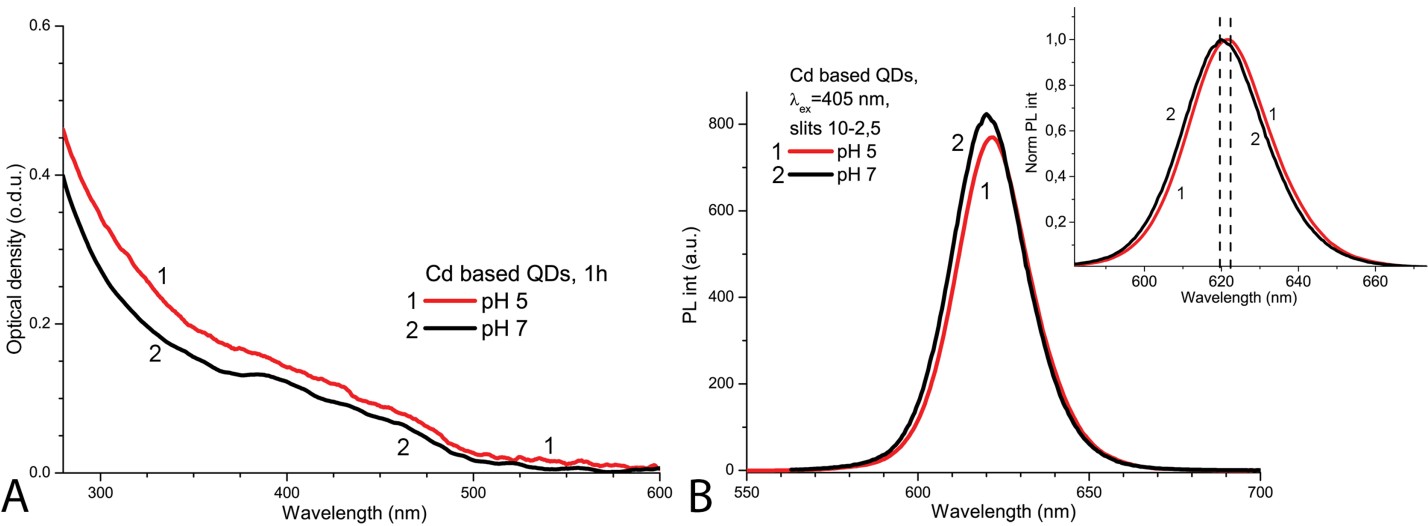

**Figure 1 The normalised spectra of Cd-based (CdSe/ZnS-COOH) QDs at pH 5.0 and 7.0 values on the first day.** The spectra of optical density (A) and the spectra of PL intensity (B). The insert shows the normalised spectra of PL. An excitation wavelength for PL was set at 405 nm, the excitation slit was 10.0 nm and the emission slit was 2.5 nm.

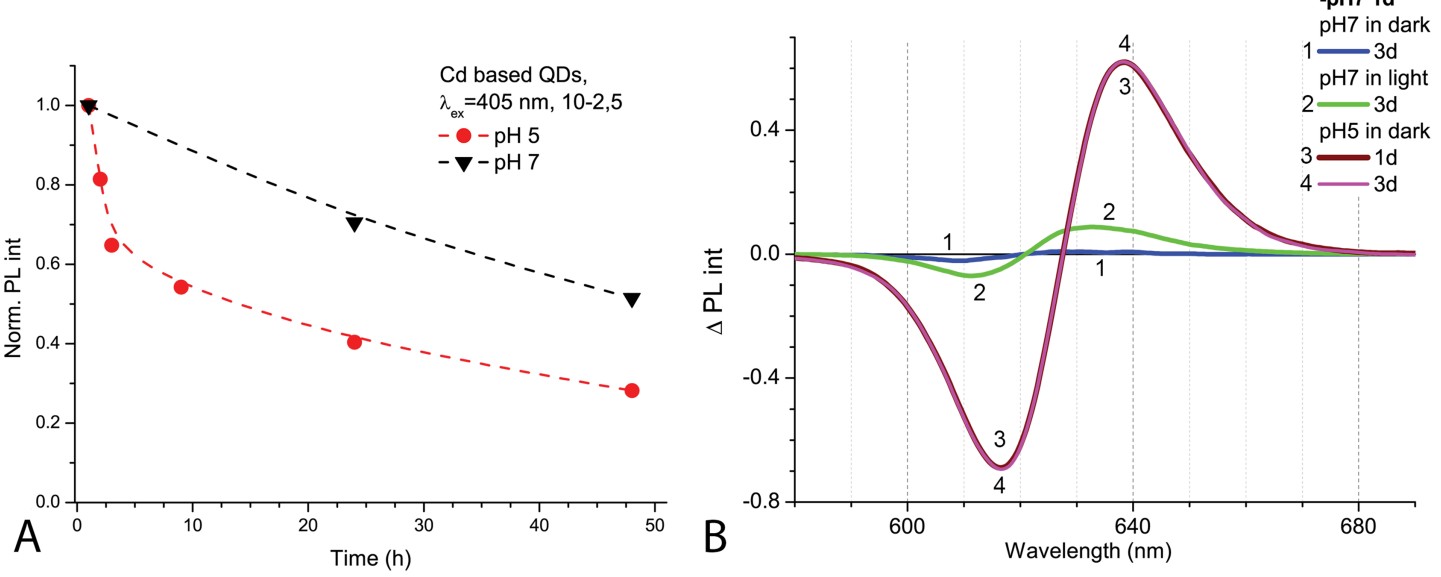

**Figure 2 The decrease of PL intensities at maximum values and the PL difference spectra of Cd-based (CdSe/ZnS-COOH) QDs.** The decrease at the PL maximum values (at about 622 nm) intensities in samples with pH values 5.0 and 7.0 over 48 h (A). The PL difference spectra of QDs that allow comparing relative spectral differences between PL changes being observed after 48 h in different pH solution with those detected at pH 7.0 in the first hour of experiment (B). An excitation wavelength for PL was set at 405 nm, the excitation slit was 10.0 nm and the emission slit was 2.5 nm.

## Bacteria identification

The 26 bacterial isolates from gut of wild *Salmo trutta* fry were identified as Gram-positive (belonged to *Carnobacterium*, *Listeria* and *Microbacterium* genera) and Gram-negative (belonged to *Aeromonas*, *Buttiauxella*, *Serratia* and *Shewanella* genera) bacteria. Sequences of isolated strains AL13; AL20; ST8; ST9; ST16 closely related to *Aeromonas*

*salmonicida* (sequences similarity 99%); AL10; ST3; ST4; ST7; ST12; ST18—to *Aeromonas sobria* (similarity 99%), AL2—to *Aeromonas popoffii* (similarity 99%), AL8; ST11; ST15—to *Aeromonas* spp. (similarity 100%), AL18; ST1; ST10; ST13; ST14—to *Shewanella putrefaciens* (similarity 99%), ST6; ST17—to *Carnobacterium maltaromaticum* (similarity 100%), ST19—to *Listeria* sp. (similarity 99%), ST32—to *Buttiauxella* sp. (similarity 99%) and ST33—to *Serratia* sp. (similarity 98%) and ST5—to *Microbacterium* sp. (similarity 99%) (Table S1).

## Antibacterial efficacy of Cd-based QDs and $Cd^{2+}$

Antibacterial activity of Cd-based QDs and $Cd^{2+}$ was tested on the 26 gut bacteria isolated from wild *Salmo trutta* fry in disk-diffusion assays. Generally, antimicrobial agent diffuses into the agar, this inhibiting germination and growth of the test microorganism and resulting in hollow zones forming around the discs (*Balouiri, Sadiki & Ibnsouda, 2016*). The agar disc-diffusion method yields qualitative interpretive category results (susceptible or resistant) and a quantitative result (zone of inhibition) (*Espinel-Ingroff, 2007*). Thus, the more susceptible to the antimicrobial activity of the compound a strain is, the larger the diameter of the inhibition zone is.

Under the experimental conditions (pH 7.0, 15 °C and 25 °C, 48 h, in light and in dark), the Cd-based QDs at the concentration of 4 nm did not show antibacterial efficacy on any of the 26 tested Gram-positive and Gram-negative gut bacteria. The experiments on the gut bacteria were repeated with Cd-based QDs at pH 5.0. The agar disc-diffusion assay showed that in the acidic medium, at 15 °C and 25 °C, Cd-based QDs (4 nm) also did not exert a growth inhibition effect on any of the 26 isolated gut bacteria. The results of these studies performed on the Gram-positive (*Listeria* sp. ST19, *C. maltaromaticum* ST17, *Microbacterium* sp. ST5) and Gram-negative (*Serratia* sp. ST33, *Aeromonas* sp. ST11, *S. putrefaciens* ST10) bacteria are presented in Figs. 3A and 3B.

As can be seen in Fig. 4 and Table S1, $Cd^{2+}$ displayed antimicrobial activities on both Gram-positive and Gram-negative gut bacteria. The resistance abilities of the isolated gut bacteria to Cd were found to widely vary, *i.e.*, the established MIC values of $Cd^{2+}$ ranged from 0.09 to 3.11 mM (Fig. 4), and the diameter values of the growth inhibition zones at these concentrations varied from 6.2 to 10.1 mm (Table S1). The obtained data showed that temperature was not a significant factor influencing the MIC value of Cd in 73% of the isolates (19 out of 26), (Fig. 4). However, the temperature was a significant factor affecting the size of the inhibition zone for some isolates at the same MIC of Cd (Table S1). The inhibition zones of six isolates *Aeromonas* sp. (AL8), *A. salmonicida* (ST9, ST16), *Buttiauxella* sp. (ST32), *S. putrefaciens* (ST 13) and *Microbacterium* sp. (ST5) were significantly larger ($F_{1.6} = 8.989$, $p = 0.024$, $F_{1.6} = 16.35$, $p = 0.007$, $F_{1.6} = 7.049$, $p = 0.038$, $F_{1.6} = 8.989$, $p = 0.024$, $F_{1.6} = 23.11$, $p = 0.003$, $F_{1.6} = 18.24$, $p = 0.005$, and $F_{1.6} = 16.89$, $p = 0.006$, respectively) at 15 °C than at 25 °C, and four isolate's *A. sobria* (ST3, ST7), *S. putrefaciens* (AL18, ST10) were significantly larger ($F_{1.6} = 95.53$, $p < 0.001$, $F_{1.6} = 34.35$, $p = 0.001$, $F_{1.6} = 8.218$, $p = 0.029$, and $F_{1.6} = 31.40$, $p = 0.001$, respectively) at 25 °C than those at 15 °C (Table S1).

Of the bacteria tested, Gram-positive *Listeria* sp. ST19 was found to be the most sensitive to $Cd^{2+}$ exposure, because the MIC values of Cd were 0.18 and 0.09 mM at 15 °C
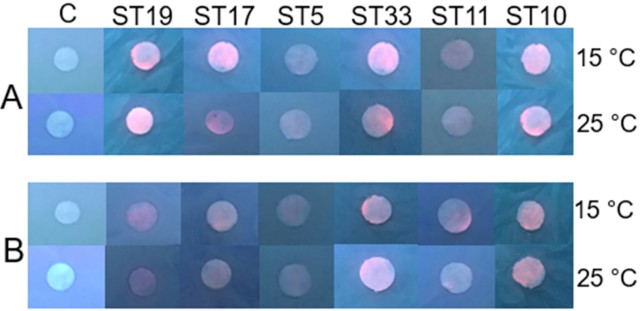

**Figure 3 Disc-diffusion assay of antibacterial efficacy of 4 nM concentrations of Cd-based (CdSe/ZnS-COOH) QDs.** Antibacterial efficacy (A) at pH 7.0; (B) at pH 5.0 on the gut Gram-positive bacteria *Listeria* sp. ST19, *C. maltaromaticum* ST17, *Microbacterium* sp. ST5, and Gram-negative bacteria *Serratia* sp. ST33, *Aeromonas* sp. ST11 and *S. putrefaciens* ST10, isolated from *Salmo trutta* fry. C, control.

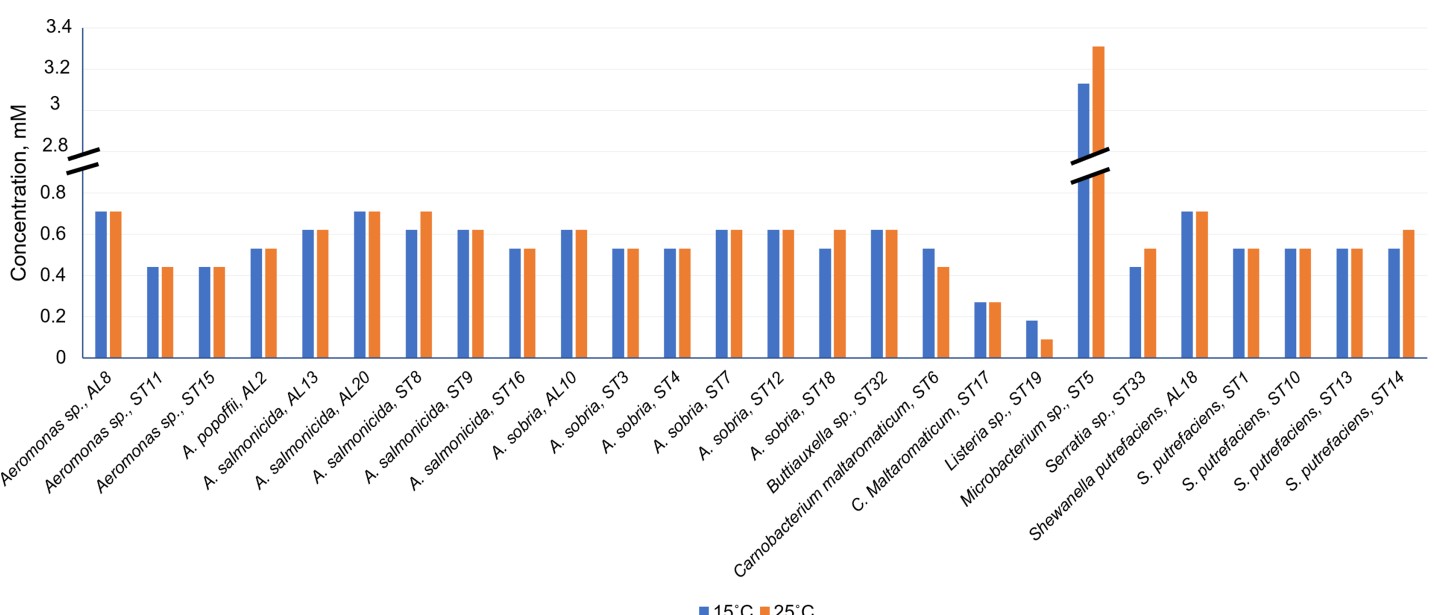

**Figure 4 Minimum inhibitory concentrations (MIC) of $Cd^{2+}$ against isolated *Salmo trutta* fry gut bacteria at 15 °C and 25 °C, pH 7.0.**

and 25 °C temperature respectively. The obtained results indicated that the inhibition zones of the *Listeria* sp. ST19 strain increased with the increasing Cd ions concentrations at 15 °C (r = 0.832) and 25 °C (r = 0.896) temperatures (Figs. 5A and S2A). The MIC value of Cd for the Gram-positive *C. maltaromaticum* ST17 strain was the same (0.27 mM) in both tested temperatures and the inhibition zones at this concentration did not statistically differ ($F_{1.6} = 0.009$, $p = 0.926$). In addition, it was noted that the size of the hollow zones around the disc exhibited increasing trend (r = 0.988 at 15 °C temperature and r = 0.978 at 25 °C temperature) with increasing $Cd^{2+}$ concentration. However, a significant difference between the inhibition zones was not observed between the two (15 °C and 25 °C) tested temperatures ($F_{4.30} = 0.281$, $p = 0.888$) (Figs. 5B and S2B).

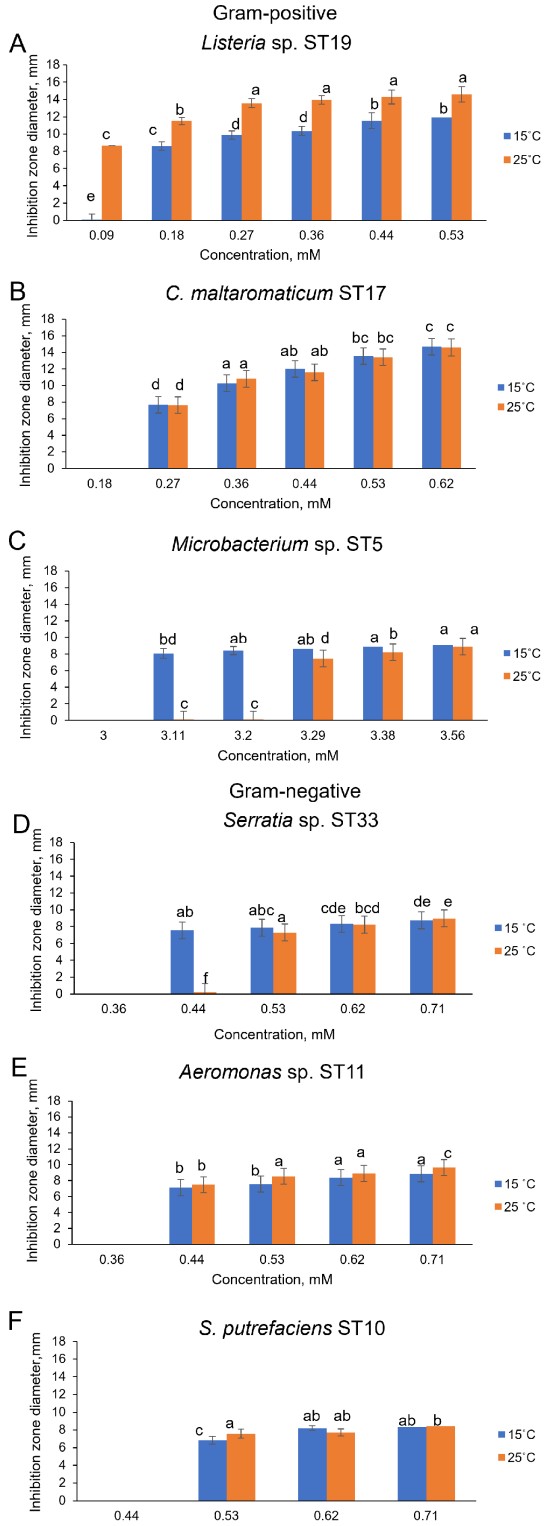

**Figure 5 Disc-diffusion assay of antibacterial efficacy of Cd²⁺.** Antibacterial efficacy on Gram-positive bacteria: *Listeria* sp. ST19 (A), *C. maltaromaticum* ST17 (B), *Microbacterium* sp. ST5 (C) and Gram-negative bacteria: *Serratia* sp. ST33 (D), *Aeromonas* sp. ST11 (E), and *S. putrefaciens* ST10 (F) at two tested temperatures (15 °C and 25 °C) at pH 7.0. a–e indicate the homogeneity of results (two-way ANOVA, $n = 4$, mean ± SD, $p > 0.05$).

The MIC values of Cd ions for the *C. maltaromaticum* ST6 strain were higher (2 and 1.7 times at 15 °C and 25 °C, respectively) than those of the ST17 strain (Table S1). The Gram-positive *Microbacterium* sp. ST5 exhibited the greatest resistance to Cd ions impact (MIC values were 3.11 mM at 15 °C and 3.29 mM at 25 °C). A significant difference between the inhibition zones was found between the two tested temperatures ($F_{4.30} = 461.36$, $p < 0.001$) (Figs. 5C and S2C). The high MIC exhibited by ST5 against $Cd^{2+}$ suggests that this strain is likely readily able to survive in an environment with high levels of cadmium contamination.

The MIC values of $Cd^{2+}$ for Gram-negative bacteria (18 out of 22 isolates) ranged from 0.44 to 0.71 mM and did not significantly differ at the tested temperatures ($p > 0.05$) (Table S1). According to the MIC values of Cd, the most sensitive among the tested Gram-negative bacteria were the *Serratia* sp. ST33 and *Aeromonas* spp. (ST11, ST15) strains (Fig. 4). A significant difference between the inhibition zones of *Serratia* sp. ST33 was found between the two tested temperatures ($F_{3.24} = 293.41$, $p < 0.001$) (Figs. 5D and S3A). The MIC values of $Cd^{2+}$ for *Aeromonas* sp. ST11 did not significantly differ between the tested temperatures ($F_{3.30} = 1.88$, $p = 0.156$) (Figs. 5E and S3B). The MIC value of $Cd^{2+}$ for the *S. putrefaciens* ST10 strain was the same (0.53 mM) in both tested temperatures, but the diameter of the inhibition zone at this concentration statistically significant increased ($F_{1.6} = 31.40$, $p = 0.001$) at 25 °C compared to 15 °C (Figs. 5F and S3C).

## Interactions of Cd-based QDs with bacteria after 9 h exposure in light

To study the interactions of Cd-based QDs with bacteria after 9 h exposure at pH 5.0, spectroscopic measurements were performed on three Gram-positive (*Listeria* sp. ST19; *C. maltaromaticum* ST19; *Microbacterium* sp. ST5) and three Gram-negative (*Serratia* sp. ST33; *Aeromonas* sp. ST11; *S. putrefaciens* ST10) bacteria (Fig. 6). Initially, the autofluorescence signals of the samples with Gram-negative bacteria were found to be stronger. There were certain specific differences observed between the strains of bacteria in both groups. The spectral position and variation of the PL bands being observed in the red spectral region at the two excitation wavelengths 400 nm (Fig. 6A) and 440 nm (Fig. 6B) can be explained by the presence of different endogenous metalloporphyrins in the bacterial samples. Thus, the signal was strongest in the case of the *S. putrefaciens* ST10 from Gram-negative and *Microbacterium* sp. ST5 from Gram-positive strains, while in the case of ST19, the formation of metal-free porphyrins was observed as well.

The PL intensity of the Cd-based QDs in the bacterial culture with supernatant (PBS, pH 5.0) after 9 h in natural light was the higher with Gram-positive bacteria than in samples with Gram-negative bacteria (Fig. 7A). After resuspension in fresh PBS (pH 5.0), the PL intensities of QDs decreased in all samples, but relatively the biggest decrease was registered in samples with *Microbacterium* sp. ST5 bacterial culture (Fig. 7B). In terms of intensities in bacterial cultures with supernatant, the fluorescence spectra of the supernatant showed the relatively highest PL intensities of QDs in *Microbacterium* sp. ST5 and *Serratia* sp. ST33 and the relatively smallest in *Listeria* sp. ST19 and *Aeromonas* sp. ST11 (Fig. 7C).

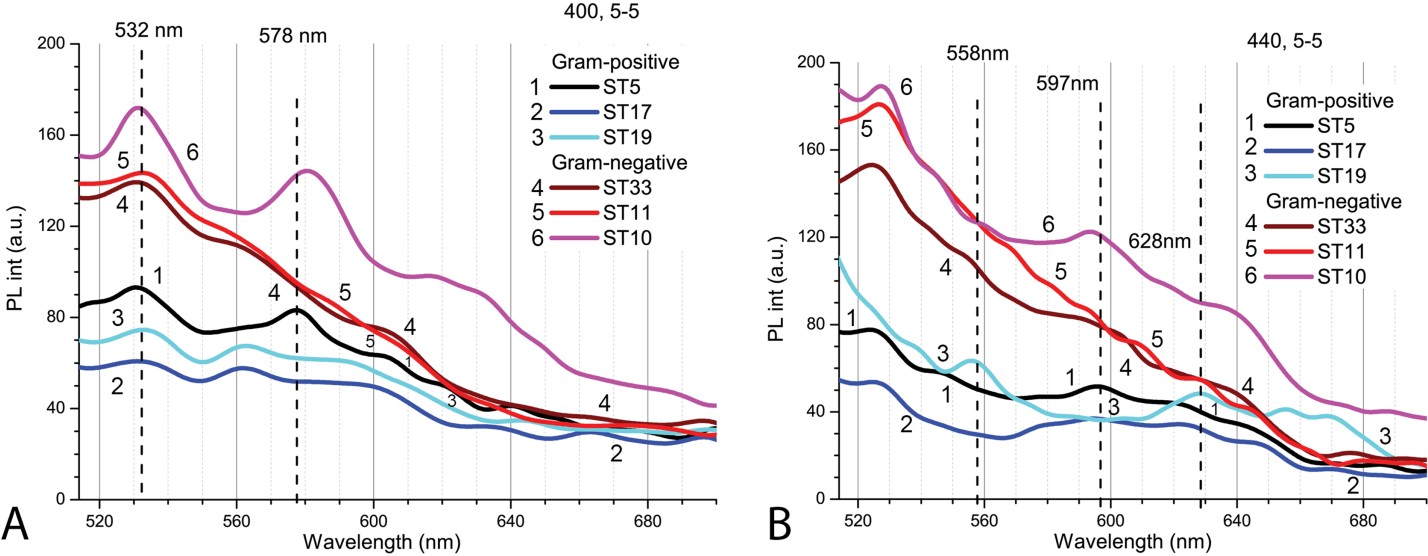

**Figure 6 The spectra of autofluorescence of three Gram-positive and three Gram-negative bacteria samples at pH 5.0 without QDs.** Gram-positive (*Listeria* sp. ST19; *C. maltaromaticum* ST19; *Microbacterium* sp. ST5) and Gram-negative (*Serratia* sp. ST33; *Aeromonas* sp. ST11; *S. putrefaciens* ST10) bacteria samples. An excitation wavelength for PL was set at 400 nm (A) and 440 nm (B), the excitation and the emission slits were 5.0 nm.

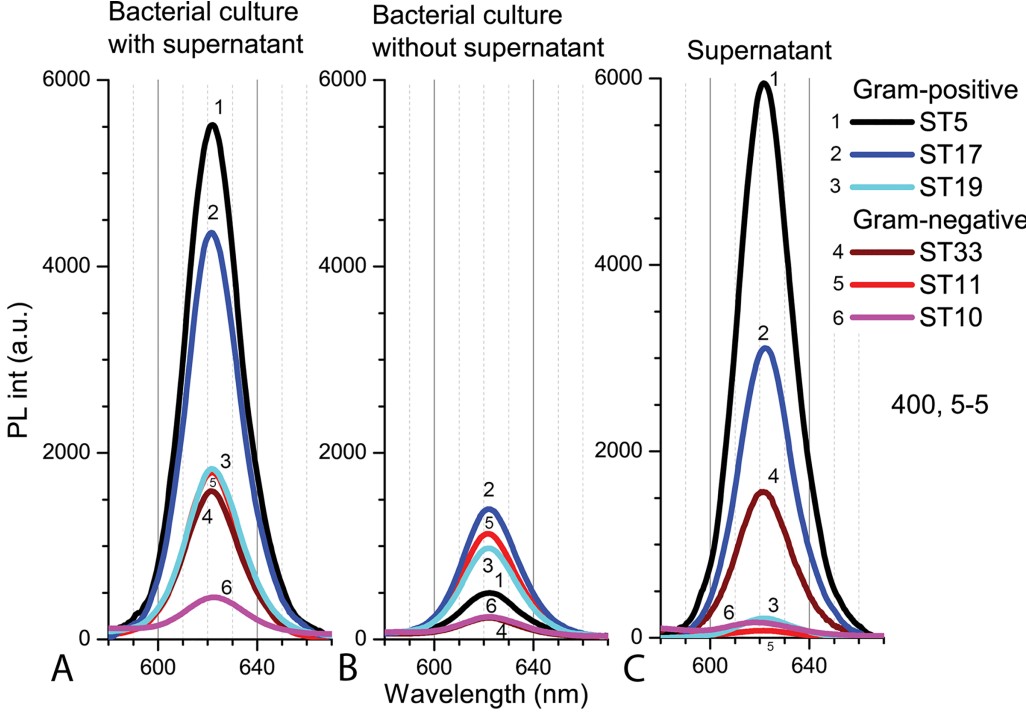

**Figure 7 The spectra of PL intensity of Cd-based (CdTe/ZnS-COOH) QDs in various media.** In bacterial culture with supernatant (A), in bacterial culture without supernatant (B) and in supernatant (C). An excitation wavelength for PL was set at 400 nm, the excitation and emission slits were 5.0 nm.

**Table 1 Effects of Cd-based (CdTe/ZnS-COOH) QDs on the viability of gut bacteria of *Salmo trutta* fry after 9 h exposure at 25 °C, pH 5.0 in light.**

| Genus/species | Control CFU/mL (mean ± SD) | CdSe/ZnS-COOH, 4 nm CFU/mL (mean ± SD) |
|---|---|---|
| *Listeria* sp. ST19 | $1.9 \pm 0.1 \times 10^8$ | $1.7 \pm 0.2 \times 10^8$ |
| *C. maltaromaticum* ST17 | $1.7 \pm 0.1 \times 10^8$ | $1.5 \pm 0.2 \times 10^8$ |
| *Microbacterium* sp. ST5 | $1.6 \pm 0.1 \times 10^8$ | $1.5 \pm 0.1 \times 10^8$ |
| *Serratia* sp. ST33 | $1.6 \pm 0.3 \times 10^8$ | $1.4 \pm 0.2 \times 10^8$ |
| *Aeromonas* sp. ST11 | $1.5 \pm 0.1 \times 10^8$ | $1.5 \pm 0.1 \times 10^8$ |
| *S. putrefaciens* ST10 | $1.7 \pm 0.4 \times 10^8$ | $1.7 \pm 0.4 \times 10^8$ |

Bacterial colony count results after exposure to Cd-based QDs in PBS (pH 5.0) are presented in Table 1. Experiments were in clear agreement with the data of the disc-diffusion assay and demonstrated that the Cd-based QDs (at the concentration of 4 nm) had insignificant effects on the viability of bacteria after 9 h exposure at 25 °C, pH 5.0 in natural light. The number of bacteria in the control samples did not significantly differ ($F_{5.24}$ = 0.362, $p$ = 0.870) from the bacteria numbers after exposure with QDs in a liquid medium (Table 1).

## DISCUSSION

External factors such as corrosive medium or prolonged exposure to light can affect the structural stability of QDs, which plays a key role in their toxicity caused by HMs leakage. It has been shown that the spectral properties of negatively charged CdTe-MSA core QDs undergo gradual changes when exposed to different doses of light in distilled water (*Kalnaitytė, Bagdonas & Rotomskis, 2018*). Similarly, the decrease of the PL intensity in acidic conditions can be explained by the detachment of the surface ligands, which give rise to defect states that lay within the band gap and reduced the quantum yield by providing alternative pathways of excited-state relaxation (*Aldana et al., 2005*). As the pH of a medium decreases, it also promotes the aggregation of QDs (*Zhang et al., 2008*) resulting in decreased intensity and a bathochromic shift of the PL peak. On the other hand, the leakage of toxic Cd ions requires serious damage to the NP structure, which is reflected by characteristic changes in the photoluminescence spectrum of QDs, such as a hypsochromic shift of the peak and a huge decrease in intensity (*Karabanovas et al., 2009*). Thus, in the absence of such spectroscopic changes compared to the initial spectra or the spectra in the control medium with neutral pH, the variation in spectral properties of the Cd-based (CdSe/ZnS-COOH) QDs at the concentration of 4 nm, observed for 48 h, presumably indicates molecular rearrangements in the coating (which can lead to subsequent aggregation of QDs) rather than structural damage to the core of the NPs and the release of Cd ions into the medium.

As at pH 7.0, the agar disc-diffusion assay showed that in the acidic medium (pH 5.0), at 15 °C and 25 °C, Cd-based QDs (4 nm) did not exert any growth inhibition effect on any of the 26 isolated gut bacteria. These results were confirmed by counting bacterial colonies

after 9 h of exposure with Cd-based QDs in a liquid medium. The Cd-based QDs at a concentration of 4 nm had no significant effects on the viability of the tested bacteria. On the other hand, the selected strains of bacteria did reveal different interaction patterns with CdSe/ZnS-COOH QDs after incubation for 9 h in natural light. The PL intensities of QDs were found to be higher in the samples with Gram-positive bacteria. Interestingly, the strains which showed the higher sensitivity towards Cd ions also revealed a relatively more intense signal of QDs PL when transferred into a fresh medium after centrifugation, this implying a stronger interaction between NPs and bacterial cells. In contrast, the biggest increase in PL intensity of QDs was detected in the supernatant in the case of the most resistant strain *Microbacterium* sp. ST5. This increase of intensity in supernatant could be explained by a lack of interactions between the NPs and bacteria exterior. The comparison between the autofluorescence spectra of the different bacterial strains registered in the control suspensions also revealed certain differences in metabolic activity. Thus, the detected fluorescence bands in the case of the most sensitive strain of Gram-positive bacteria indicated the presence of endogenous metal-free porphyrins. The spectral bands of the most resistant strain revealed a notable intensity of metalloporphyrins, the enhanced production of which and subsequent release might serve as protective means to avoid potential oxidative damage.

There is considerable debate about the effects of cadmium on microbial growth. Cd-based QDs can be ingested by aquatic organisms and released Cd ions may effect endogenous microbial communities. Since comprehensive knowledge and data regarding the toxicity of Cd-based QDs on fish gut microbiota still limited, it was important also to evaluate the effect of Cd on the isolated gut bacteria. As it has been reported, Cd present in food can cause health risks even at low doses, for instance, it could cause disruption of mice gut microbiota (*Liu et al., 2014, 2020*). It was found that HMs exposure alters the composition and metabolic profile of the gut microbiota at the functional level and, in turn, the gut microbiota alters the uptake and metabolism of metals (*Duan et al., 2020*; *Zhang et al., 2020*; *Zhai et al., 2017*). A study performed using Nile tilapia (*Oreochromis niloticus*) exposed over 4 weeks to water contaminated with 1 mg/L of Cd emphasized a deep change in gut microbial communities (*Zhai et al., 2017*). Other studies have demonstrated that concentrations (2; 5 and 10 mg/L) of $Cd^{2+}$ at 72 h exposure could alter the richness, diversity and composition of microbiota in the intestine of crayfish *Procambarus clarkii*. *Shewanella* species were elevated in Cd-treated groups compared with a control. By contrast, *Buttiauxella* sp. showed significantly lower abundances after exposure to $Cd^{2+}$ (*Zhang et al., 2020*). As other authors point out, Gram-positive bacteria can be more susceptible to toxicity posed by HMs and NPs than Gram-negative bacteria because they lack an outer cell membrane with lipopolysaccharide (LPS) chains that play a protective role (*Fazeli, Hassanzade & Alae, 2011*; *Feng et al., 2015*; *Pramanik et al., 2018*).

Our study revealed the antibacterial effect of $Cd^{2+}$ on isolated gut bacteria from wild *Salmo trutta* fry. Gram-positive bacteria *Listeria* sp. ST19 and *C. maltaromaticum* ST17 were found to be the most sensitive to $Cd^{2+}$. By contrast, the Gram-positive *Microbacterium* sp. ST5 exhibited the greatest resistance to Cd ions of all tested bacteria.

This may be related to the extrusion mechanism that enables Gram-positive bacteria to pump Cd ions out (*Bhakta et al., 2012*). Efflux pumps are the most prevalent bacterial tools conferring HMs resistance (*Bazzi et al., 2020*). For most isolates, the MIC of Cd ions was independent of temperature, but some bacterial strains were more sensitive at lower temperatures. This can be explained by the fact that the gut bacteria were isolated from the wild *S. trutta* and salmonids are a cold-water fish species (*Elliot, 1991*; *Guillemette et al., 2011*). Thus, some of gut bacteria of salmonids are more adapted to lower temperatures, resulting in a faster response to Cd ions at lower temperatures. The susceptibility of gut bacteria to Cd ions was also assessed by the size of the zones of inhibition. The size of the inhibition zone mostly differed according to the type of bacteria, to the concentrations of Cd ions and to temperature. With potential consequences for host population responses to climate change, temperature variation shapes the composition and function of animal gut microbiomes, key regulators of host physiology. Although gut microbiota is characterized by plasticity to responses to temperature due to the existence of conserved mechanisms, extreme temperatures can disrupt the stability of diversity within the gut microbiota (*Sepulveda & Moeller, 2020*).

## CONCLUSIONS

The Cd-based (CdSe/ZnS-COOH) QDs at a concentration of 4 nm were found to be stable in aqueous media (with pH 7.0) or undergoing superficial structural changes favoring the formation of aggregates (at pH 5.0), so did not apparently release HMs into the media over 48 h in conditions of light or dark and did not produce an inhibitory effect on the growth and viability of the gut bacteria isolated from wild *Salmo trutta* fry after short-term (9 h and 48 h) incubation. $Cd^{2+}$ exerted significant dose-dependent toxic effects on the isolated gut bacteria, and the size of the inhibition zone varied significantly between the temperatures tested in some of isolates. The most sensitive and the most resistant to $Cd^{2+}$ were Gram-positive bacteria, with their MIC values differing significantly between two tested temperatures, while the MIC values for most of the tested Gram-negative bacteria showed no such variation. Comparison of the PL spectra revealed an interesting trend: among the selected Gram-positive and Gram-negative strains, those with the higher sensitivity towards $Cd^{2+}$ also revealed relatively stronger signals of QDs PL when transferred after incubation into a fresh medium. In addition, the formation of endogenous metalloporphyrins observed spectroscopically in some bacterial strains indicates certain differences in metabolic activity that may play a protective role against potential oxidative damage. Our *in vitro* study on the isolated fish gut bacteria showed that the possible release of Cd from QDs could also induce remarkable changes in the structure and functions of the fish gut microbiota *in vivo* by causing dysbiosis due to $Cd^{2+}$ antibacterial efficacy. In our study, we used a limited concentration of QDs and specific environmental conditions, these may be insufficient for ruling out Cd-based (CdSe/ZnS-COOH) QDs toxicity. Further studies are expected to enable the predicting of potential health risks posed by Cd-based QDs to aquatic organisms.

### Funding

This work was supported by the Research Council of Lithuania, Project No. S-MIP-20-22. The funders had no role in study design, data collection and analysis, decision to publish, or preparation of the manuscript.

### Grant Disclosures

The following grant information was disclosed by the authors:
Research Council of Lithuania: S-MIP-20-22.

### Competing Interests

The authors declare that they have no competing interests.

### Author Contributions

- Renata Butrimienė performed the experiments, analyzed the data, prepared figures and/or tables, authored or reviewed drafts of the article, and approved the final draft.
- Agnė Kalnaitytė performed the experiments, analyzed the data, prepared figures and/or tables, authored or reviewed drafts of the article, and approved the final draft.
- Emilija Januškaitė performed the experiments, analyzed the data, prepared figures and/or tables, and approved the final draft.
- Saulius Bagdonas conceived and designed the experiments, performed the experiments, analyzed the data, prepared figures and/or tables, authored or reviewed drafts of the article, and approved the final draft.
- Živilė Jurgelėnė conceived and designed the experiments, performed the experiments, analyzed the data, prepared figures and/or tables, authored or reviewed drafts of the article, and approved the final draft.
- Dalius Butkauskas performed the experiments, analyzed the data, prepared figures and/or tables, molecular identification of gut bacteria, and approved the final draft.
- Tomas Virbickas performed the experiments, analyzed the data, prepared figures and/or tables, and approved the final draft.
- Danguolė Montvydienė performed the experiments, prepared figures and/or tables, and approved the final draft.
- Nijolė Kazlauskienė conceived and designed the experiments, analyzed the data, authored or reviewed drafts of the article, and approved the final draft.
- Vesta Skrodenytė-Arbačiauskienė conceived and designed the experiments, performed the experiments, analyzed the data, prepared figures and/or tables, authored or reviewed drafts of the article, and approved the final draft.

### Animal Ethics

The following information was supplied relating to ethical approvals (*i.e.*, approving body and any reference numbers):

A permit to perform sampling of *Salmo trutta* fry specimens in the Kena River, Permit No. 025, was issued in 2020 to the Nature Research Centre by the Environmental Protection Agency under the Ministry of the Environment of the Republic of Lithuania.

## Field Study Permissions

The following information was supplied relating to field study approvals (*i.e.*, approving body and any reference numbers):

Permit No. 025, was issued in 2020 to the Nature Research Centre by the Environmental Protection Agency under the Ministry of the Environment of the Republic of Lithuania.

## DNA Deposition

The following information was supplied regarding the deposition of DNA sequences:

GenBank accession numbers MW441229 to MW441233; OL739143 to OL739153.

## Data Availability

The raw data are available in the Supplemental Files.

## Supplemental Information

Supplemental information for this article can be found online at http://dx.doi.org/10.7717/peerj.14025#supplemental-information.

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
