# Peer review of "Interactions of semiconductor Cd-based quantum dots and Cd2+ with gut bacteria isolated from wild Salmo trutta fry"

_PeerJ, doi:10.7717/peerj.14025_

## Round 0.1 · original submission · Major Revisions

Please address the comments of the three Reviewers and revise the manuscript according to the suggestions.

Reviewer 1 ·

Basic reporting

no comment

Experimental design

no comment

Validity of the findings

no comment

Additional comments

In this manuscript Interactions of Semiconductor Cd-based Quantum Dots and Cd2+ with gut bacteria isolated from wild Salmo trutta fry, the authors use spectroscopic method to study the interactions between quantum dots CdSe/ZnS and gut bacteria. The experiment is interesting, and its design is rigorous. I think there are some issues should be addressed before publication.

1. In Fiugre 1, the spectra of absorption and emission are displayed for CdSe/ZnS-COOH QDs. The PL spectrum is normal. However, the exciton feature/peak in absorption is not found around 600nm (J. Am. Chem. Soc. 2005, 127, 8, 2496–2504). I checked the raw data, where 500-1100nm is almost flat. Such large emission Stokes shift for QDs should be explained.

2. At low pH with the surface ligand -COOH, CdSe/ZnS-COOH aggregates easily. To confirm the aggregation, TEM (transmission electron microscopy) or DLS (dynamic light scattering) (Nanoscale, 2019, 11, 34–44) can be applied to directly show the QD clusters. DLS can provide information of hydrodynamic diameter, which can provide evidence of the detachment of the surface ligands (line 343)

3. In Figure 6, the inhibition zones’ circularity is not that perfect. The authors chose diameter instead of area. Is there any specific reason? How were diameters measured by ImageJ?

4. Minor: What is the meaning of HMs? Heavy metals?

Overall, the topic is interdisciplinary, interesting and essential. This experiment has been well performed with rational design and detailed discussion. I recommend publishing after dealing with the above concerns.

Reviewer 2 ·

Basic reporting

The current ms discusses the spectral properties and antibacterial activities of CdSe QDs. The manuscript is beautifully written, but the novelty of the ms is shallow, as there are several works published on the interactions of NPs to aquatic bacteria.

Experimental design

In line #57, more recent references may be used for the biological application of QDs.
Line # 84-85 is misleading because the toxicity of QDs depends on functionalization. Core-shell QDs are not toxic and do not induce any antibacterial properties to bacteria. They have been used as sensors to monitor the movement of bacteria. This has been discussed in Arshad et al RSC Adv. 2016, 6.
The disc diffusion method used here for antibacterial activity would not be an appropriate method for QDs. There are two possibilities, which the author may define 1) The QDs may aggregate in PBS and may not be available at the expected concentration 2) the QDs with their large size compared to Cd2+ might be trapped in the disc or may not diffuse into the agar medium to show an antibacterial property. The authors would have used liquid assays to test antibacterial activity. The fluorescence around the disc also would be a better tool to confirm the diffusion of QDs.

Validity of the findings

Line #338, more studies are needed to confirm the release of Cd2+ from QDs. The interaction of QDs with bacteria also includes the possibility of attachment to the surface of bacteria. Arshad et al (2016) reported the charge-based interactions of QDs with bacterial surfaces and possibilities of internalization and toxicities. What is the chance of different pH solutions interfering with this charge-based interaction if it happens in environmental bacteria?
Line #351-371: It is possible that the shell of QDs can be digested by gastric enzymes and low pH and core particles become toxic. But how do you confirm this degradation based on spectral properties alone? Authors may need to do additional microscopic analysis (TEM/AFM) to substantiate this claim.

Reviewer 3 ·

Basic reporting

1. The introduction is well written. I would recommend the authors to briefly rationalize in the introduction why specific experimental conditions were chosen. E.g., why pH 5.0 and 7.0? Why not test a more acidic or a more basic environment? Why 4nM instead of higher or lower? What is the reported or expected concentration range of QDs in rivers and lakes contaminated by NPs?
2. Line 101: “the aim of this study was to investigate the spectroscopic properties of commercially available Cd-based QDs..” The rationale behind conducting spectroscopy studies is unclear for the readers, especially in the introduction. Perhaps it would be helpful to i) briefly explain their hypothesis on how emission properties of Cd-QDs may correlate with their toxicity to microbes in the introduction, and then explain in detail in the spectroscopic properties sub-section of the Results section.
3. Structure-wise, I found that the discussion section contains rationales that could be briefly stated in the introduction or/and in the results section. This will help readers to better understand the methods chosen by the authors for this work.

Experimental design

4. Methods were well written. I suggest the following improvements, which may require additional experiments to be done or/and literature search:
o If both pH and salinity levels are important (as the authors have cited Rocha et al.), have the authors considered to vary salinity levels for their biological testing?
o Have the authors considered to test different concentrations of QDs? Is this what Figure 5 is showing? If yes, why in the Introduction and in the Method section the authors only mentioned concentration=4nM?

Validity of the findings

5. Rationale could be further improved
o Lines 220-238: the authors described and showed spectroscopic measurements of QDs at different pH values without providing an explanation of why a difference was observed. Would it be helpful to briefly rationalize these results right after describing them, perhaps based on existing literature?
6. underlying data analysis could be further improved
o Figure 5: would it be helpful to label based on “mmole/L” or “nmole/L” instead of “mg/L”, if the hypothesis is that Cd2+ is responsible for antibacterial activities? The mass of Cd-QDs can vary based on the type of QDs used, but the concentration (in moles) of Cd could stay the same.
o Discussion lines 337-342: very clear rationale here. I recommend the authors to briefly explain how the emission profile of QDs correlates to the release and Cd in a sentence or two in the introduction and again in the spectroscopy subsection of the Results.
o Lines 351-358 described the reason behind choosing 4nM of Cd QD for this study. This should be briefly explained (in 1-2 sentences) in the introduction and/or in the results section.

Additional comments

7. Figures 1 is perhaps supporting information - it can either be moved to SI or be combined with figure 2 to be more concise.
8. Figure 6 is very poor in resolution – please use original high-resolution figures

---

## Round 0.2 · accepted · Accept

Thank you for carefully considering the Reviewer comments and revising your manuscript accordingly. The manuscript can now be accepted for publication.

Reviewer 1 ·

Basic reporting

no comment

Experimental design

no comment

Validity of the findings

no comment

Additional comments

The authors have well-addressed my comments. Here is an additional comment to Comment 3 (reviewer 1). If the circularity is not perfect, it is better to measure the area first and calculate the diameter accordingly instead of measuring several times of the diameter.

Reviewer 3 ·

Basic reporting

The authors have addressed all comments.

Experimental design

The authors have addressed all comments.

Validity of the findings

The authors have addressed all comments.